# Learning Feasible Causal Algorithmic Recourse: A Prior Structural Knowledge Free Approach

## Abstract

Algorithmic recourse (AR) has made significant progress by identifying small perturbations in input features that can alter predictions, which provide a data-centric approach to understand decisions from diverse black-box models on the Web. Towards the feasibility issue, i.e., whether the recoursed examples provides actionable and reliable recommendations to end-users, causal algorithmic recourse have incorporated structural causal model (SCM) to preserve the realistic constraints among input features. For instance, preserving structural causal knowledge between "age" and "educational level" can avoid generating samples with decreasing age and increasing educational level. However, previous causal AR methods suffer from the requirement of prior **structural causal knowledge**, e.g., prior causal graph or the whole SCM, which restricts the realistic application of causal AR methods.

To bridge this gap, we aim to develop a novel framework for causal algorithmic recourse that does not rely on neither prior causal graph or prior SCM. Since identifying counterfactuals without causal graph is impossible, we instead propose to approximate and constrain the variation of the perturbed components, i.e., the exogenous noise variables, by formulating the generation of AR as the structure-preserving intervention. With the aid of development in non-linear Independent Component Analysis (ICA), our method can further achieve theoretically guaranteed constraints on such variation of exogenous variables. Experimental results on synthetic, semi-synthetic, and real-world data demonstrate the effectiveness of our proposed methods without any prior causal graph or SCM knowledge.[1]

## CCS Concepts

• **Computing methodologies** → **Machine learning**; • **Social and professional topics** → **User characteristics**.

## Keywords

Causality, Algorithmic Recourse, User Recommendation

---

[1]**Relevance statement**: This paper is closely relevant to the topic of *Explainable and interpretable methods for personalization* in the *User modeling, personalization and recommendation* Track in the Web Conference. Our research focuses on a web-specific interpretable area that provide user-friendly recourse towards intelligent decisions from black-box Machine Learning models on the Web. Contributions include the breakthrough towards the feasibility issue in algorithmic recourse, a prior structure-free recommendation approach, the corresponding theoretical guarantee, and extensive experimental validations.

---

**ACM Reference Format:**
Anonymous Author(s). 2018. Learning Feasible Causal Algorithmic Recourse: A Prior Structural Knowledge Free Approach. In *Proceedings of Make sure to enter the correct conference title from your rights confirmation emai (Conference acronym 'XX).* ACM, New York, NY, USA, 13 pages. https://doi.org/XXXXXXX.XXXXXXX

## 1 Introduction

The abundance of web data creates opportunities to enhance decision-making on the Internet, e.g., loan approval on the Web, online employment matching, and online medical consultation [29, 33, 44, 56, 59]. Accompanied with the wide deployment of machine learning (ML) models for data-driven and automatic decision-making, explaining complex decisions from black-box ML models is of increasing importance to safeguard the rights of end-users [24, 34]. In response to such requirement, algorithmic recourse has gained popularity in recent years by modifying input features to change model predictions [54]. For instance, a bank provides online loan service with algorithmic recourse to inform loan applicants of actions that would lead to approval [29, 44, 56]. Serving as a popular, data-centric explanation paradigm, algorithmic recourse provides incentive and user-friendly approach to help end-users to understand diverse decisions from online ML models [21, 24, 31].

One fundamental challenge for algorithmic recourse is to generate feasible real-world examples [24], i.e., providing actionable suggestions for end-users. To be specific, feasibility refers to preserving realistic constraints among input features. Despite providing insight into black-box ML models on the Web, current algorithmic recourse often fails to offer feasible recommendations for individual users. For instance, suggesting to increase in education level and a decrease in age for loan approval is meaningless to end-users, as shown in Figure 1(b).

Towards the issue of feasibility, researchers have pointed out that introducing causality [36] can benefit preserving realistic relationships among input variables [31, 34]. With the bank loan example in mind, preserving functional relationships in the Structural Causal Model (SCM), e.g., Educational_level = $f$(Age_level) ($f$ is an non-decreasing function), coincides with the target of feasible algorithmic recourse that preserves the non-decreasing relationship between the education variable and the age variable.

To this end, eailer recourse methods regulate explanation generation by constraining the distance between generated samples and SCM-derived samples when the underlying SCM is known [31]. Furthermore, in the case of unknown SCM, [24] assumes the accessibility of prior causal graph, and proposes to identify SCMs using the Gaussian process or Conditional Variational Encoder (CVAE). However, we argue that either the full SCM or the causal graph knowledge is often limited in realistic cases [5, 6]. In fact, the causal discovery task and SCM identification task are specifically proposed to identify such prior knowledge [9, 26].

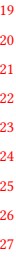
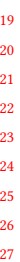

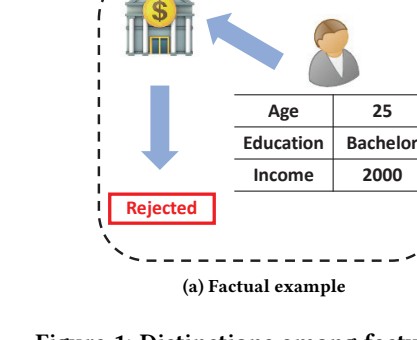

(a) Factual example

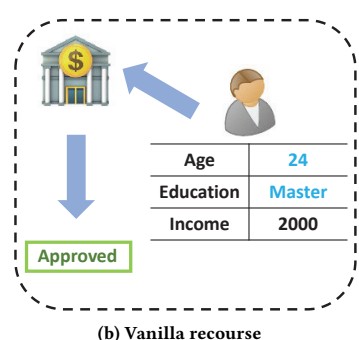

(b) Vanilla recourse

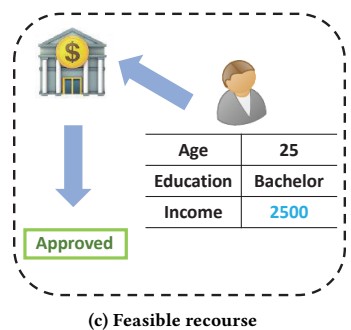

(c) Feasible recourse

**Figure 1: Distinctions among factual examples rejected by the ML decision model, vanilla algorithmic recourse that alter predictions but are not feasible, and feasible algorithmic recourse complying with realistic causal relationships.**

To bridge the gap between causally-inspired recourse and prior-absent realistic data environment, this paper proposes the **first** feasible algorithmic recourse framework without the support of prior causal graphs or SCMs. However, the critical challenge of our target arises from the fact that identifying exact counterfactuals, i.e., recourse samples, from observational data solely is nearly impossible [15, 24, 46]. To overcome this challenge, we instead theoretically re-formulate the algorithmic recourse process as a structure-preserving (SP) intervention operation [16, 53], and then approximates counterfactuals directly from observational data with the aid of advances in the area of non-linear Independent Component Analysis (ICA) [14, 15]². To ensure the feasibility, we finally constrains the divergence between generated samples to their approximated counterfactuals by constraining the variation of exogenous noise during the SP intervention. The contributions of this paper are summarized as follows:

• By reformulating the algorithmic recourse as a structure preserving intervention process, we are the pineeror research to explore the feasibility issue of algorithmic recourse without causal prior knowledge.

• Theoretically, we show that exogenous noise can be identified in a reliable manner by constructing an exogenous regressor. Subsequently, we further prove that the variation of the exogenous noise is governed by that of representations learned by the exogenous regressor under mild conditions.

• Practically, we propose two practical methods, AR-L2 and AR-Nuc, which constrain the magnitude and sparsity of variations in exogenous representations, respectively. Extensive experimental results verify that our methods: (a) significantly improve the preservation of causal relationships for algorithmic recourse; (b) successfully achieve the alteration of predictions with little cost.

## 2 Related Work

**Algorithmic Recourse.** Traditional local explanation methods for black-box ML models on data, such as tabular data, are crucial for users to interpret decisions [41]. However, these explanations often differ from complex ML models on the Web. Algorithmic

recourse (or counterfactual explanation) offers consistent and interpretable examples as an alternative for understanding decisions from online ML models [24, 31, 54]. They can be categorized into gradient-based methods [24, 31, 32, 54] and methods based on linear programming [21, 51]. Recent discussions have also addressed fairness, transportability, and reliability issues in algorithmic recourse [1, 52].

While the definition of algorithmic recourse has parallels with adversarial examples [3, 43], the **biggest** distinction between the two directions is that the former only aims to explain to the model while the latter aims to suggest both *interpretable and feasible recommendations* to end-users [22]. Unfortunately, most of the current algorithmic recourse methods lack such capability since they usually ignore the relationships and constraints among the input features [24, 51].

**Causality for Feasible Algorithmic Recourse.** To overcome the above-mentioned challenge, several recent works have suggested the incorporation of causality into the algorithmic recourse [21, 24, 31, 51]. From the view of causality, the infeasibility of vanilla algorithmic recourse stems from the fact that such recourse are generated by independently manipulating each input feature. As a consequence, the causal relationship/constraints are broken during the generation process (as shown in Figure 1(b)). Accessing the entire SCM model, [31] suggests regularization of algorithmic recourse by minimizing differences between perturbed features and their parent-generated counterparts. Building on this concept, several works[23, 24, 52] seek optimal intervention feature sets with minimal cost. These methods relax the requirement from knowing the whole SCM model to knowing the causal graph. For instance,[24] proposes approximating the SCM model using Gaussian process or Conditional Variational Encoder(CVAE). Moreover, [21] explores different cost setups.

**Structural-Preserving Intervention.** In line with traditional counterfactual theory [36], recent studies have developed alternative framework termed as *backtracking counterfactuals* to describe counterfactuals with tractable formulations [4, 16, 53]. The core difference between traditional counterfactuals and backtracking counterfactuals stands on the interpretation of interventions. To be specific, backtracking counterfactuals proposed the **Structural-Preserving Intervention (SP Intervention)** as modifications on

---

²To be specific, it means that algorithmic recourse can be formulated as the process which solely manipulates the exogenous noise of each sample, while the structural causal relationships among features remain.

exogenous noise variables in a structural causal model, i.e., the functional relationships among variables are preserved. Such formulations are divergent from traditional counterfactuals, where both the functional relationships are modified [36] (see Appendix C for details). Inspired from backtracking counterfactuals, we model algorithmic recourse as the process of SP interventions [16], as such formulations is coherent to the algorithmic recourse [36].

## 3 Background

**Notations.** We consider a binary classifier $h$ as the underlying ML model [21, 24, 31], while our method also allows for a multi-categorical classifier. $h$ is deployed on a dataset $\mathcal{D}$ with $M$ samples, each consisting of $n$-dimensional input characteristics $\mathbf{x}$ and output labels $y$. We index data points as $1 \leq i \leq M$ and features as $1 \leq j \leq n$. The goal of algorithmic recourse (AR) is to answer why some individuals were denied loans and what actions they could take to increase approval chances. We search for the closest algorithmic recourse $\mathbf{x}^R$ for a given factual sample $\mathbf{x}^F$ using model $\mathcal{M}$: $\mathbf{x}^R \in \arg\min_{\mathbf{x} \in \mathcal{X}} \mathrm{d}\left(\mathbf{x}, \mathbf{x}^F\right)$ s.t. $h(\mathbf{x}) \neq y$, where $\mathrm{d}$ refers to some predefined distance functions. The counterfactual $\mathbf{x}^R$ is often computed as $\mathbf{x}^R = \mathbf{x}^F + \delta$ by adding some perturbations $\delta$ on $\mathbf{x}^F$.

**Feasibility of AR.** Most prior work on algorithmic recourse considers features as independently manipulable inputs, which ignores the potentially rich causal structure over $\mathbf{x}^F$. In recent years, a bunch of authors [21, 23, 24, 31] have argued for the feasibility of AR, i.e., the need to consider causal relations between variables during the process of recourse.

**Structural Causal Models (SCMs) and Causal Graph.** From the perspective of causality, the data-generating process of $\mathbf{X}$ is described by a Markov/semi-Markov structural causal model (SCM) [36] $\mathcal{V} = (\mathbf{X}, \mathbb{F}, P_\sigma)$ describes the causal relations between $n$ features in $\mathbf{X} = \{\mathbf{x}_1, \mathbf{x}_2, \ldots, \mathbf{x}_n\}$ as: $\mathbb{F} = \left\{f_j \mid \mathbf{x}_j := f_j(\mathbf{x}_{\mathrm{pa}(j)}, \sigma_j)\right\}_{j=1}^n$, where $\sigma$ denotes exogenous noise variables with distributions as $P_\sigma$, and $\mathbb{F}$ is the set of assignment functions $f_j$ which maps the causal parents $\mathbf{x}_{\mathrm{pa}(j)}$ to each variable $\mathbf{x}_j$. Following previous protocols [24, 31], we here assume the non-existence of unmeasured confounders, i.e., the *causal sufficiency* assumption [36], such that the exogenous noise $\sigma_j$ are mutually independent. Meanwhile, the SCM is often coupled with its intuitive illustration, i.e., the causal graph $G$, which is formed by a one-to-one mapping from each variable $\mathbf{x}_j$ to a node in $G$ and directed edges drawn from $\mathbf{x}_j$ to $\mathbf{x}_{\mathrm{pa}(j)}$ for $j \in [n]$. We assume that $G$ [24] is *acyclic* such that the coupled SCM is non-recursive. By recursively resolving the parents in terms of their parents [4, 53], $\mathbf{x}$ can be represented in the form of exogenous variables: $\mathbf{x} = f(\sigma)$, where $f : \mathbb{R}^n \mapsto \mathbb{R}^n$ can be regarded as the *exogenous-aggregated* version of $\{f_j\}_{j=1}^n$ (see Appendix D for details). Besides, we use the notation $\mathbb{F} \circ \mathbb{F}_A^{-1}$ to denote the three steps including abduction, action, and prediction to compute counterfactuals w.r.t some intervention set $A$ [23, 24].

**Interventions with Minimal Cost.** Assuming the SCM model $\mathcal{V}$ is accessible with invertible forms, e.g., additive SCMs, [23] formulate the above algorithmic recourse problem as finding the

**Table 1: Comparison on the requirement of different Prior Knowledge across different settings.**

| Research | [23, 31] | [24] | Ours |
|---|---|---|---|
| Prior Causal Graph | ✔ | ✔ | ✘ |
| Prior SCM | ✔ | ✘ | ✘ |

optimal intervention strategy with minimal cost:

$$\mathbf{A}^* \in \arg\min_{\mathbf{A} \in \mathcal{V}} \mathrm{cost}(\mathbf{A}; \mathbf{x}^F)$$
$$\text{s.t. } \mathbf{x}^R = \mathbb{F} \circ \mathbb{F}_A^{-1}\left(\mathbf{x}^F\right), \quad h(\mathbf{x}^R) \neq h(\mathbf{x}^F), \quad (1)$$

where $\mathrm{cost}(\mathbf{A}; \mathbf{x}^F)$ is the cost function measures how intervention set $A$ varies $\mathbf{x}^F$, and $\mathbf{A}^*$ directly specifies the set of feasible actions to be performed for minimally costly recourse. By the three steps of structural counterfactuals [36], the counterfactual examples, i.e., $\mathbf{x}^R$, is generated based on the evidence $\mathbf{x}^F$ and $A$ [23]. Based on this foundation, some relaxation has been proposed in [24] by assuming only the access of causal graph $G$ rather than the entire SCM $\mathcal{V}$. Unfortunately, as the ultimate goal of causal discovery [61], the prior causal graph still restricts the application of algorithmic recourse. Hence, how to maintain the causal relationship without prior knowledge of causal graph and SCM is crucial in many scenarios.

**Connections between ICA and Causal Inference.** ICA aims to identify *mutually independent* source signal $S$ from mixed observations $T$ via a mixture function $\tilde{f}: T = \tilde{f}(S)$, e.g. the cocktail party problem [13]. While the traditional ICA theory can successfully identify the non-Gaussian distributions of $\tilde{f}$ and $\sigma$ under the condition that $\tilde{f}$ is a linear function [46], practical applications often involve non-linear functions for $\tilde{f}$, making it extremely challenging to directly infer $\tilde{f}$ without additional information [14]. Previous advances in causal inference have utilized ICA to identify different forms of SCMs [10, 46]. For instance, the identification of causal graph $G$ for linear and additive SCMs, i.e., $\mathbf{x}_j := w_j \mathbf{x}_{\mathrm{pa}(j)} + b_j$ for $j \in [n]$ ($w, b$ are linear coefficients), can be reformulated into a linear ICA problem [46, 61]. On the other side, [10] has contributed novel identification results for non-linear ICA based on independent causal mechanisms of source signals. These works build up the foundation of our work to identify and constrain the exogenous noise for learning feasible algorithmic recourse.

## 4 Methods

### 4.1 Prior Assumptions, Problem and Clarification

**Assumptions and Problem.** We put no restrictions on the form of the underlying data generation mechanism, i.e., SCM of $\mathbf{x}$: $\mathbf{x}_j := f_j(\mathbf{x}_{\mathrm{pa}(j)}, \sigma_j)$. We only assume the **causal sufficiency** of observed $\mathbf{x}$. Our target is to preserve the causal structures over $\mathbf{x}^F$, i.e., $\mathbb{F} = \{f_j\}_{j=1}^n$, during the recourse process from $\mathbf{x}^F$ to $\mathbf{x}^R$ such that features of $\mathbf{x}^R$ follow the feasibility principle (shown in Fig. 1(c)). To the best of our knowledge, our paper is the pioneer work to generate feasible algorithmic recourse without prior structural knowledge.

**Clarifications.** We clarify that different from [24], we do not model the recoursed sample $\mathbf{x}^R$ as traditional counterfactuals (with hard interventions), but rather model $\mathbf{x}^R$ as a tractable, special counterfactuals named backtracking counterfactuals [53] by structure-preserving (SP) interventions [4]. To be specific, prior researches treat the recoursed sample $\mathbf{x}^R$ as the result of the hard intervention on $\mathbf{x}^F$. For instance, assuming the set of intervention values as $A$, then $\mathbf{x}^R_j := A_j$ for intervented variable $j$, while $\mathbf{x}^R_k := f_j(\mathbf{x}^R_{\mathrm{pa}(k)}, \sigma_k)$ for those non-intervented variable $k$. However, due to the lack of both causal graph and SCM in our problem, such formulation is not tractable to identify the variation of $f$ during the process of recourse. Instead, we define the recoursed sample $\mathbf{x}^R$ as the result of SP interventions that do not vary structural functions $\{f_j\}_{j=1}^n$ but only manipulate exogenous noise variables $\{\sigma_j\}_{j=1}^n$. Combined with the exogenous-aggregated representation $\mathbf{x} = f(\sigma)$ stated in Sec.2, the resulting recoursed sample and the factual sample can be written as:[3]

$$\mathbf{x}^R = f(\sigma^R), \quad \mathbf{x}^F = f(\sigma^F). \tag{2}$$

*Remark* 4.1. We note that the exogenous-aggregated form in (2) and (2) implies that the structural relationship, i.e., $f$, remains invariant during recourse, which is coherent to the feasibility goal.

### 4.2 Identification of the Exogenous Noise

**Challenge Statement.** However, as one can only access to $\mathbf{x}^R, \mathbf{x}^F$ without any knowledge on $f, \sigma^F, \sigma^R$, it is impossible to identify $f$ and directly enforces identified $f$ to be invariant[4]. Alternatively, we propose that identifying another core term in (2) and (2), i.e., the exogenous variables $\sigma$, and constraining the variation of identified quantities should be beneficial to the preservation of $f$. Essentially, significant variability in $\sigma$ closely resembles point-wise intervention. To identify such variations of $\sigma$, we resort to advances of non-linear ICA [15].

**Relating Algorithmic Recourse to ICA.** As we have assumed the causal sufficiency, i.e., Markov SCMs, the exogenous noise elements $\sigma$ are mutually independent [36]. In an analog, the fundamental requirement of ICA is that source signals should be independent from each other. Consequently, the exogenous-aggregated formulation in (2) has natural connections to the formulation of the non-linear ICA [14], i.e., $T = f(S)$, where the signal $\sigma^R$ can be interpreted as the source signal $S$, the aggregated function $f$ represents the mixing process $\tilde{f}$, and $\mathbf{x}^R$ represents the mixed observations $T$.

**Identification of the exogenous noise.** According to the theory in [15], we can identify the exogenous noise to some degree by introducing an observed auxiliary variable. Such auxiliary variables could be historical observations, the time variables or the class labels [14]. Specifically, we choose output label $y$ as the auxiliary variable in this paper. Following the identifying strategy in [15], we randomly permute the sample order of $y$ to eliminate correlations with $\sigma$ and construct the modified data $\mathcal{D}^A$ using permuted $y$. Finally, we employ a discriminative model called the **exogenous**

---

[3]We note that traditional counterfactuals with hard interventions cannot achieve such forms.

[4]We note that although some advances in learning SCM from observational data are present [35], while some strict parametric constraints should be followed, e.g., additive non-linear model (ANM).

**regressor** to distinguish between $\mathcal{D}^A$ and $\mathcal{D}$ through a non-linear regression system as follows:

$$\min_\theta \sum_{i=1}^M l\left(\frac{1}{1+\exp(-r(\mathbf{x}_i, y_i))}, o_i\right) \quad s.t. \quad r(\mathbf{x}, y) = \sum_{j=1}^n \psi_j^\theta\left(\phi_j^\theta(\mathbf{x}), y\right), \tag{3}$$

where the binary labels $\mathbf{o}$ indicate the source of the data as either $\mathcal{D}$ or $\mathcal{D}^A$. The functions $\psi_j^\theta : \mathbb{R}^2 \mapsto \mathbb{R}$ and $\phi_j^\theta : \mathbb{R}^n \mapsto \mathbb{R}$ are non-linear representations parameterized by $\theta$, implemented using deep networks [15]. In this paper, we refer to $\phi$ as the **exogenous representations** for simplicity. We then offer theoretical insights into the behavior of the learned $\phi_\theta(\mathbf{x})$ to support the validity of our exogenous regressor as follows:

**Theorem 4.2** (Identification of $\sigma$). *Assume that:*
  (a) *The exogenous noise $\sigma$ is conditionally exponential of order $K$ of $y$.*
  (b) *There exists $nk+1$ realizations of $y$ as $\{y\}_{l=0}^{nk}$ such that the difference matrix consists of $\{y\}_{l=0}^{nk}$ with some transformations is invertible.*
  (c) *The trained (deep) logistic regression system in (3) has the universal approximation capability to distinguish $\mathcal{D}$ from $\mathcal{D}^A$.*

*Then, in the case of infinite samples, the representations $\phi^\theta(\mathbf{x})$ identifies $\sigma$ up to a linear transformation of point-wise statistics $\tilde{\mathbf{q}}$:*

$$\tilde{\mathbf{q}}(\sigma) = \mathbf{A}\phi_\theta(\mathbf{x}) + \mathbf{b}, \tag{4}$$

*where $\mathbf{A}$ and $\mathbf{b}$ are fixed but unknown matrices.*

Notably, although the above theorem provides the general case for any $k \geq 1$, we will only treat the cases when $k = 1$ throughout the following parts.

### 4.3 Constraining the variation of the exogenous noise

Although the exogenous representations $\phi_\theta(\mathbf{x})$ learned in (16) does not directly identify $\sigma$, we show that constraining the variation of $\phi_\theta(\mathbf{x})$ still governs that of $\sigma$. We first establish a connection between the variation of exogenous representations $\phi_\theta(\mathbf{x})$ and exogenous noise $\sigma$ as follows:

$$\mathbf{A}\left(\phi(\mathbf{x}^F) - \phi(\mathbf{x}^R)\right) = \tilde{\mathbf{q}}(\sigma^F) - \tilde{\mathbf{q}}(\sigma^R). \tag{5}$$

We then construct the variation of $\phi_\theta(\mathbf{x})$ on a batch of samples as $\mathbf{H}$:

$$\mathbf{H} = \{\phi(\mathbf{x}_1^F) - \phi(\mathbf{x}_1^R), \ldots, \phi(\mathbf{x}_{M_b}^F) - \phi(\mathbf{x}_{M_b}^R)\}, \tag{6}$$

where $M_b$ refers to the batch sample size. Consequently, we further demonstrate that constraining the sparsity and magnitude of $\mathbf{H}$ adequately restricts the corresponding characteristics of $\sigma^F - \sigma^R$, respectively. For the clarity to distinguish $\mathbf{H}$ from $\mathbf{H}^0$, we have:

$$\begin{cases} \mathbf{H}^0 = \{\sigma_1^F - \sigma_1^{CF}, \sigma_2^F - \sigma_2^{CF}, \ldots, \sigma_{M_b}^F - \sigma_{M_b}^{CF}\}, \\ \mathbf{H} = \{\phi(\mathbf{x}_1^F) - \phi(\mathbf{x}_1^{CF}), \phi(\mathbf{x}_2^F) - \phi(\mathbf{x}_2^{CF}), \ldots, \phi(\mathbf{x}_{M_b}^F) - \phi(\mathbf{x}_{M_b}^{CF})\}, \end{cases}$$

As one can not access to the original variation of the exogeneous noise, i.e. $\mathbf{H}^0$, we then propose two strategies to design indirect constraints on $\mathbf{H}^0$ via the accessible and learnable $\mathbf{H}$. We note that all designed optimization process only updates the generated $\mathbf{X}^\mathbf{R}$ while keeping $\phi$ and factual input $\mathbf{x}^F$ fixed.

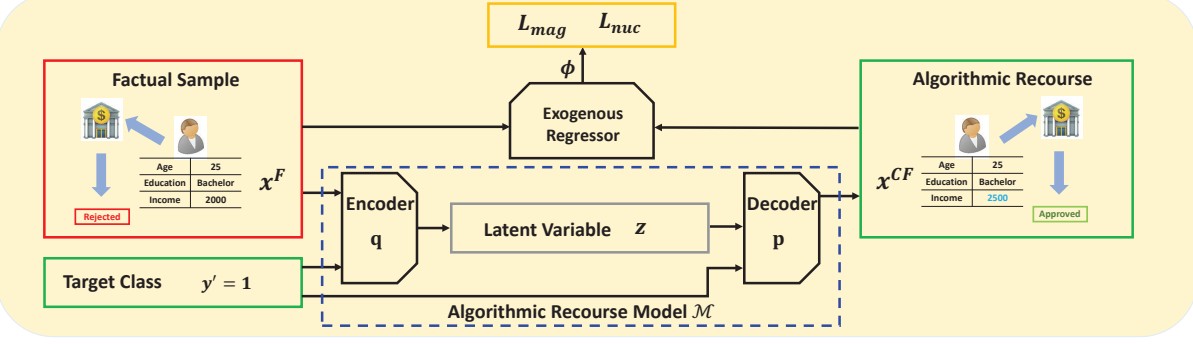

**Figure 2: Overall illustration of our framework, where exogenous regressor constrains the generation of the algorithmic recourse model.**

**Sparsity Constraint.** The intuition behind the sparsity constraint is that we expect the number of perturbed input features to be small. To this end, we propose the algorithmic recourse with Nuclear Norm (AR-Nuc) by optimizing the nuclear norm of $\mathbf{H}$ as $\|\mathbf{H}\|_*$:

$$\min_{\mathbf{X}^R} : \mathcal{L}_{\text{nuc}} = \|\mathbf{H}\|_*.$$

To theoretically justify the validness of our proposed AR-Nuc, we show that the sparsity of $\mathbf{H}$, i.e., $\mathcal{L}_0(\mathbf{H})$, is further constrained by the nu-clear norm of $H$:

**Theorem 4.3** (Connection between $\mathbf{H}$ and $\mathbf{H}^0$). *Assuming that the transformed exogeneous noise is bounded, i.e., $\epsilon \leq \|H_{ij}^\sigma\|$, the sparsity of $\mathbf{H}^0$ is governed by that the nu-clear norm $\mathbf{H}$:*

$$\mathcal{L}_0(H^0) \leq \frac{cn^{3/2}}{\epsilon} \mathcal{L}_{nuc}(H), \tag{7}$$

*where c is the element-wise upper-bound of A.*

**Magnitude Constraint.** Beyond the sparsity constraint, we restrict the exogenous noise to vary in a small magnitude as well. To this end, we design another method named the algorithmic recourse with $\mathcal{L}2$ norm (AR-L2). More specifically, we first provide the following theorem to argue that optimizing the $\mathcal{L}2$ norm of the variation of exogenous representations, $\phi(\mathbf{x}^F) - \phi(\mathbf{x}^R)$, is enough to constrain that of $\sigma^F - \sigma^R$:

**Theorem 4.4.** *Assume the sufficient statistics $\tilde{\mathbf{q}}$ (k=1) is a bi-Lipschitz function of $\sigma$, then $\|\sigma^F - \sigma^R\|_2$ is governed by $\|\phi(\mathbf{x}^F) - \phi(\mathbf{x}^R)\|_2$, where $\|\cdot\|_2$ is the $\mathcal{L}2$ norm.*

We then propose the algorithmic recourse with $\mathcal{L}2$ Norm (AR-L2) by optimizing the $\mathcal{L}2$ norm of $\mathbf{H}$ on $\min_{\mathbf{X}^R} : \mathcal{L}_{\text{mag}} = \|\mathbf{H}\|_2$. (see Appendix F for proofs).

### 4.4 Choice of Recourse Backbone

As the conditional variational autoencoder (CVAE) provides a flexible and reliable approach [31, 34], we adopt the previous proposed CFVAE model [31] as the algorithmic recourse model $\mathcal{M}$ in this paper. More specifically, we achieve this by maximizing the log-likelihood of $P(\mathbf{x}^R \mid \mathbf{x}^F, y')$, where $y'$ refers to the target prediction altered from the original decision $y$. Following previous protocol

[31], we instead maximize the evidence lower bound (ELBO) of $P(\mathbf{x}^R \mid \mathbf{x}^F, y')$ by following:

$$\mathbb{E}_{q(\mathbf{z}|\mathbf{x}^F, y')} \log p(\mathbf{x}^R \mid \mathbf{z}, y', \mathbf{x}^F) - \mathcal{KL}\left( q(\mathbf{z} \mid \mathbf{x}^F, y') \parallel p(\mathbf{z} \mid y', \mathbf{x}^F) \right). \tag{8}$$

where we first arrive the latent representations $\mathbf{z}$ via the encoder $q(\mathbf{z} \mid \mathbf{x}^F, y')$ and then generate the counterfactual $\mathbf{x}^R$ via the decoder $p(\mathbf{x}^R \mid \mathbf{z}, y', \mathbf{x}^F)$. Meanwhile, the prior conditional density of $\mathbf{z}$ is sampled from a normal distribution: $p(\mathbf{z} \mid y', \mathbf{x}^F) \sim N(\mu_{y'}, \sigma_{y'}^2)$ to achieve a closed form of the KL-divergence. For realizations, we adopt the $\mathcal{L}1$ norm to measure the reconstruction loss, with an additional Hinge loss to force the ML model $h$ to alter the prediction from $y$ to $\mathbf{y}'$:

$$\mathcal{L}_{\text{recon}}(\mathbf{x}^F, \mathbf{x}^R) = \log p\left(\mathbf{x}^R \mid \mathbf{z}, y', \mathbf{x}^F\right) = \|\mathbf{x}^R - \mathbf{x}^F\|_1,$$

$$\mathcal{L}_{\text{hinge}}(h(\mathbf{x}^R), y', \beta) = \max(h_y(\mathbf{x}^R) - h_{y'}(\mathbf{x}^R), -\beta),$$

where $h_y(\mathbf{x}^R)$ refers to the predicted score (e.g., a probability in $[0, 1]$) from $h$ at class $y$, $\beta$ is the hyper-parameter to control the margin. Finally, by performing the monte-carlo approximation and sampling from the encoder $q(\mathbf{z} \mid \mathbf{x}^F, y')$, we express the original loss for optimizing $\mathcal{M}$ on a batch sample with size $M_b$ as follows:

$$\mathcal{L}_{\text{ori}} = \sum_{i=1}^{M_b} \mathcal{L}_{\text{recon}} + \mathcal{L}_{\text{hinge}} + \mathcal{KL}(y_i', \mathbf{z}_i, \mathbf{x}_i^F), \tag{9}$$

where $\mathcal{KL}(y_i', \mathbf{z}_i, \mathbf{x}_i^F)$ refers to the empirical estimation of:

$$\mathcal{KL}\left( q(\mathbf{z} \mid \mathbf{x}^F, y') \parallel p(\mathbf{z} \mid y', \mathbf{x}^F) \right).$$

*Overall Loss.* $\mathcal{L}_{\text{nuc}}$ and $\mathcal{L}_{\text{mag}}$ can be incorporated into the above objective to preserve the causal relationships. Therefore, the overall objective function can be written as $\mathcal{L}_{\text{ori}} + \alpha_{\text{nuc}} \mathcal{L}_{\text{nuc}}$ and $\mathcal{L}_{\text{ori}} + \alpha_{\text{mag}} \mathcal{L}_{\text{mag}}$, where $\alpha_{\text{nuc}}$ and $\alpha_{\text{mag}}$ are hyper-parameters (see Appendix E for details).

*Remark* 4.5 (Manipulation Issue). Besides the feasibility issue, another critical issue of causal algorithmic recourse, i.e., the manipulation issue, should be satisfied during the recourse process. More formally, manipulation requires that the generated sample will not change variables which are under protection, e.g., gender or race.

**Table 2: Results of the distribution and proximity scores on synthetic and German Loan data: Metrics are Mean±STD over 5 repeated experiments, with the best Dist_score highlighted.**

| Setting | | In-sample | | Out-of-sample | |
|---|---|---|---|---|---|
| Metric | | Proximity | D-Score | Proximity | D-Score |
| **Benchmark: Synthetic** | | | | | |
| Vanilla | CEM | 4.82 ± 0.85 | -369.74 ± 8.9 | 3.79 ± 0.62 | -372.50 ± 10.2 |
| | CFVAE | 2.12 ± 0.51 | 2.31 ± 0.26 | 2.09 ± 0.55 | 2.30 ± 0.25 |
| Partial | CFA-a | 2.24 ± 0.07 | -4.76 ± 2.10 | - | - |
| | CFA-p | 2.18 ± 0.11 | -2.53 ± 1.15 | - | - |
| Ours | AR-Nuc | 2.38 ± 0.26 | **3.26 ± 0.28** | 2.37 ± 0.15 | **3.08 ± 0.22** |
| | AR-L2 | 2.06 ± 0.44 | **3.03 ± 0.12** | 2.07 ± 0.22 | **3.12 ± 0.05** |
| *Oracle* | AR-SCM | *2.11 ± 0.32* | *3.58 ± 0.21* | *2.28 ± 0.27* | *3.66 ± 0.08* |
| **Benchmark: German** | | | | | |
| Vanilla | CEM | 4.67 ± 0.51 | 0.68 ± 0.27 | 4.67 ± 0.44 | 0.49 ± 0.25 |
| | CFVAE | 6.14 ± 0.13 | 1.02 ± 0.14 | 6.15 ± 0.15 | 1.03 ± 0.10 |
| Partial | CFA-a | 6.04 ± 0.20 | 0.99 ± 0.05 | - | - |
| | CFA-p | 6.10 ± 0.18 | 0.83 ± 0.19 | - | - |
| Ours | AR-Nuc | 5.95 ± 0.14 | **3.42 ± 0.10** | 5.80 ± 0.13 | **3.45 ± 0.13** |
| | AR-L2 | 6.02 ± 0.10 | **3.35 ± 0.08** | 6.01 ± 0.11 | **3.40 ± 0.07** |
| *Oracle* | AR-SCM | *6.18 ± 0.27* | *3.49 ± 0.17* | *6.19 ± 0.26* | *3.51 ± 0.09* |

We argue that different kinds of interventions, i.e., point intervention, soft intervention, or structure-preserving intervention, happen on different levels, while manipulation only becomes meaningful on $X$. Hence, when considering the issue of manipulation, we only have to add constraints on the input-output reconstruction loss of the underlying VAE backbone, i.e., enforcing the corresponding feature not to be changed during the recourse.

*Remark* 4.6 (Implicitly Estimation of Counterfactual Components). We would like to clarify that our method provides implicitly estimation of counterfactuals, i.e., through quantifying and constraining the variation of exogenous variables, to achieve feasible AR. Instead of estimating full SCM and designing constraints, we focus on the exogenous components.

## 5   Experiments

In this section, we first introduce the baselines we compared, together with the evaluation metrics. Then we provide experimental results on a synthetic dataset, a semi-synthetic dataset, and a real-world dataset. Notably, our experimental data are *not* generated from linear SCMs.

**Baselines with their implementations.** Our baselines can be divided into three levels:

- Vanilla algorithmic recourse methods without any prior knowledge. Such methods include (1) the CFVAE model we introduced before [31] and (2) the CEM model, which models the perturbation using the auto-encoder structure [7];
- Recourse methods with partial prior causal knowledge. We choose the minimal-intervention framework proposed in [24] with two instantiations: (1) CFA-a: Which allows all

input features can be intervened when computing counterfactuals; (2) CFA-p: Which allows partial features can be intervened when computing counterfactuals. To be specific, both CFA-a and CFA-p requires the full graph as input, and deploy conditional variational encoders to estimate the underlying SCM;
- Oracle baselines, which refers to the methods with the whole SCM model as a prior. Such a method is implemented on the basis of the CFVAE regularized by the causal distance proposed in [31], which we call the AR-SCM method.

We implement the CFA method in two versions: CFA-All (CFA-a) and CFA-Partial (CFA-p), allowing interventions on all features or only a subset, respectively (see Appendix G for details). Meanwhile, we note that **both CFA-a and CFA-p methods do not support out-of-sample validation, as their recourse process performs separately on each sample rather than a training-then-testing paradigm [23].** On the real-world dataset, due to the lack of prior causal graph, we refer to the CFA method with all nodes allowed to be intervened as CFA-Discover, as we pre-train a causal discovery model [61] to learn the prior causal graph from data in prior.

**Details on Implementation of our models** We implemented the CFVAE algorithmic recourse model in our work. The encoder has two branches: one for estimating $\mu_{y'}$ and another for $\sigma^2 y'$. Both branches have 5 MLP layers with ELU activation and batch normalization, but $\sigma^2 y'$ uses a Sigmoid function in the final layer to constrain variance. The decoder mirrors this structure. We use BCE loss for training since the domain label is binary. The black-box model $h$ consists of 2 MLP layers with ELU activations. Our methods (AR-Nuc and AR-L2) share the same architecture and training settings as CFVAE. For regression, we use a neural network with

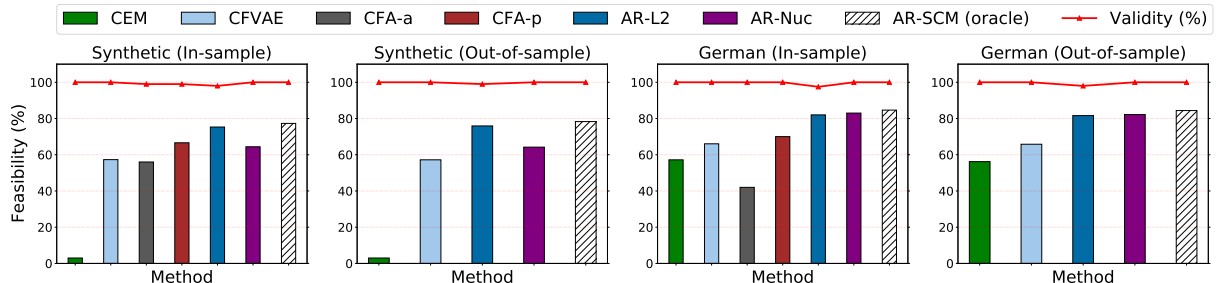

Figure 3: Feasibility and valid scores for the synthetic data and the German Loan data.

Table 3: High-dimensional Performance, where Time is the running time for recoursing per sample.

| Methods | | Proximity | D-Score | Validity (%) | Feasibility (%) | Time (s) |
|---|---|---|---|---|---|---|
| Vanailla | CEM | 5.61±1.24 | <-500 | 99 | <10 | - |
| | CFVAE | 2.68±0.88 | 1.52±0.36 | 100 | 35.6 | 0.12 |
| Partial | CFA-a | 3.21±0.04 | -5.37±2.28 | 100 | 34.7 | Over 1h |
| | CFA-p | 2.72±0.13 | -2.62±1.93 | 100 | 52.8 | Over 1h |
| Ours | AR-Nuc | 2.54±0.30 | 3.61±0.30 | 100 | 73.1 | 0.15 |
| | AR-L2 | **1.97±0.14** | **3.38±0.08** | 100 | **78.4** | **0.14** |
| *Oracle* | CF-SCM | *2.28±0.35* | *3.78±0.11* | *99* | *82.0* | *0.17* |

three MLP layers and ReLU activation, optimized with Adam (learning rate = 0.001). Hyperparameters $\beta$, $\alpha_n$, and $\alpha_m$ are set to 0.2, 2, and 2, respectively, for all experiments.

**Metric.** We evaluate the quality of generated algorithmic recourse using the following metrics [24, 31]:

- **Feasibility Score** (%): Percentage of individuals whose algorithmic recourse satisfy the prior monotonic constraints, indicating feasibility;
- **Distribution Score**: Log-likelihood ratio of generated algorithmic recourse compared to the given causal edges, indicating compliance with the SCM model, which equals to $\log p(\mathbf{x}_j^R \mid \mathbf{x}_{Pa(j)}^R)/\log p(\mathbf{x}_j^F \mid \mathbf{x}_{Pa(j)}^F)$;
- **Validity** (%): Percentage of individuals with favorable predictions from algorithmic recourse;
- **Proximity**: Average $\mathcal{L}$1-distance between counterfactual and original features for continuous features, and number of mismatches for categorical features.

Intuitively, the validity score and the Proximity score are commonly adopted metrics to measure how AR methods achieves target accuracy with little cost [21, 34, 54]. Meanwhile, Feasibility Score and Distribution Score are common metrics to quantify how each AR method achieves feasibility principle [24, 31]. We note that these four metrics exists widely in previous studies. We conduct experiments and compute metrics in two settings: in-sample, testing the model on training samples, and out-of-sample, testing on samples outside the training dataset without output labels. In our experiments, we mainly answer two questions:

- *How does our method perform on preserving the causal relationship?*

- *Does our method sacrifice other metric (e.g., the Proximity or Validity) to improve the feasibility?*

**Synthetic dataset.** We simulate a synthetic dataset with three features as $(\mathbf{x}_1, \mathbf{x}_2, \mathbf{x}_3)$ and one outcome variable $(y)$. To incorporate a monotonically increasing causal relationship between $\mathbf{x}_1, \mathbf{x}_2$ and $\mathbf{x}_3$, we follow [31] to adopt the structural equations:

$$\mathbf{x}_1 \sim N(\mu_1, \sigma_1), \quad \mathbf{x}_2 \sim N(\mu_2, \sigma_2)$$
$$\mathbf{x}_3 \sim N(k_1 * (\mathbf{x}_1 + \mathbf{x}_2)^2 + b_1, \sigma_3), \quad (10)$$
$$y \sim \text{Bernoulli}(k_2 * (\mathbf{x}_1 * \mathbf{x}_2) + b_2 - \mathbf{x}_3),$$

where we set $\mu_1 = \mu_2 = 50$, $\sigma_1 = 15$, $\sigma_2 = 17$, $\sigma_3 = 0.5$, $k_1 = 0.0003$, $k_2 = 0.0013$, and $b_1 = b_2 = 10$ as in [31]. Obviously, the causal relationship embodied in this dataset is $\mathbf{x}_1, \mathbf{x}_2$ increase $\Rightarrow \mathbf{x}_3$ increases; and $\mathbf{x}_1, \mathbf{x}_2$ decrease $\Rightarrow \mathbf{x}_3$ decreases. Thus the feasibility set $C$ equals to the above two constraints. For method CFA-a, we allow $\mathbf{x}_1, \mathbf{x}_2$ and $\mathbf{x}_3$ to be intervened, while only $\mathbf{x}_1$ and $\mathbf{x}_2$ are allowed to be intervened for CFA-p.

Table 2 and Figure 3 demonstrate the effectiveness of our method, AR-Nuc and AR-L2. It achieves significant improvements in the feasibility and distribution scores. Compared to the vanilla CFVAE, our feasibility score improves by over 15%. AR-Nuc and AR-L2 perform competitively with the ground truth approach (AR-SCM) on feasibility and distribution scores. Notably, our methods outperform CFA-a and CFA-p, even with prior causal graph.

**German Loan Dataset.** A semi-synthetic dataset including 7 variables called "German Loan" was created based on the German Credit UCI dataset [24] (see appendix for the causal graph with SCMs). For the German Loan dataset, the CFA-p method was implemented with non-interventive features (age, gender, and duration), and a constraint set (C) was used to measure feasibility, following

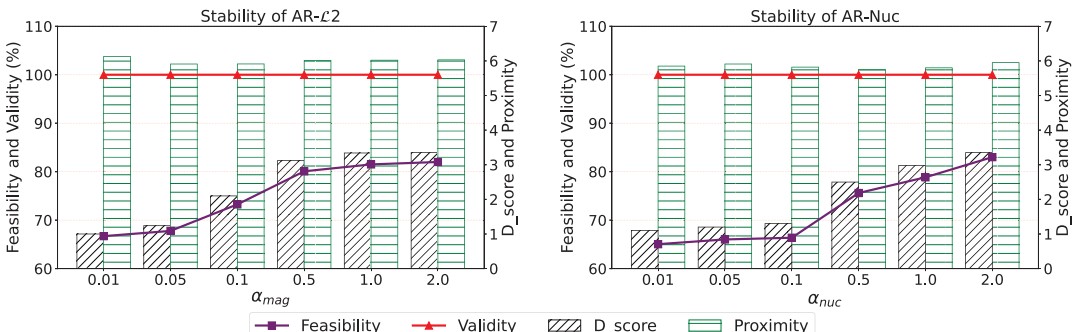

**Figure 4: Stability of of our methods across different hyper-parameter selection protocols.**

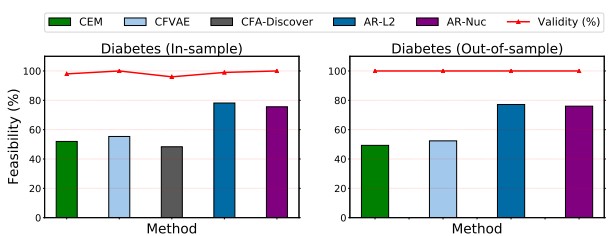

**Figure 5: Feasibility and validity scores on Diabetes.**

three rules: (1) Loan amount (L) increases $\Rightarrow$ loan duration (D) increases; (2) Age (A) increases $\Rightarrow$ income (I) increases; (3) A increases $\Rightarrow$ education-level (E) increases. As shown in Table 2 and Figure 3, our AR-Nuc and AR-L2 outperform others in feasibility and distribution scores while maintaining 100% validity at a low proximity cost.

**Diabetes Dataset.** The Diabetes dataset [21], collected by Smith [47], consists of medical records for Pima Indians with and without diabetes. It includes 8 input features such as pregnant status, blood pressure, with the class label indicating diabetic conditions. To discover the causal structure, we use the CFA method [24] with the NOTEARS method [60] (https://github.com/xunzheng/notears). Based on prior knowledge [21, 47], the constraint set $C$ includes three rules: (1) Blood Pressure $\rightarrow$ BMI, (2) Glucose $\rightarrow$ BMI, and (3) Skin thickness $\rightarrow$ BMI.

As shown in Table 4 and Figure 5, real-world experimental results entail that our approaches effectively constrain the variation of the exogenous noise, which further preserves the effect of the structural functions in generated examples. Besides, the first-discover-then-generation approach is difficult for realistic cases, as the error of discovery and approximation will accumulate together.

**Towards High-Dimensional Data.** We test the high dimensional capability of our method by involving a synthetic dataset with 80 features in our study (see Appendix (G) for details). As shown in Table 3, our methods, AR-Nuc and AR-L2, offer improved scalability in high-dimensional settings. By contrast, the need to consider every possible subset of the total feature set for conducting interventions in CFA-a and CFA-p results in exponential complexity relative to the total feature set.

**Table 4: Proximity score of the Diabetes dataset.**

| Setting | | In-sample | Out-of-sample |
|---|---|---|---|
| Vanilla | CEM | 7.42±0.11 | 7.43±0.08 |
| | CFVAE | 16.49±0.52 | 16.19±0.47 |
| Partial | CFA-Discover | 6.67±0.26 | – |
| Ours | AR-Nuc | 6.43±0.18 | 6.40±0.16 |
| | AR-L2 | 6.48±0.19 | 6.50±0.11 |

**Ablations and Parameter Stability.** We note that ablation studies are already provided by comparing CFVAE with out AR-Nuc and AR-L2. Finally, we have tested the stability of our methods, AR-L2 and AR-Nuc, by varying the hyper-parameters $\alpha_{mag}$ and $\alpha_{nuc}$. The in-sample prediction results in Fig 4 show that (a) our methods have weak effects on the feasibility when $\alpha \leq 0.1$; (b) our AR-L2 and AR-Nuc does not ruin other metrics such as proximity when improving the feasibility; (c) the feasibility achieved by our methods does not rely on the sophisticated tuning of hyper-parameters $\alpha_{mag}$ and $\alpha_{nuc}$ (only require the hyper-parameter not to be too small).

## 6 Discussion and Future Work

**Conclusion.** To protect the vulnerable end-users toward the decision models, we enhance the feasibility of algorithmic recourse such that the users can obtain both interpretable and feasible recommendations. We achieve this by identifying and constraining the variability of the exogenous noise. Extensive experimental results have verified the effectiveness of our methods.

**Limitations and Future Work.** However, one limitation remains to be addressed in future work, as our method assumes *causal sufficiency* with no unobserved features. Such a case might exist in a wide of real-world scenes, and previous work including ours [23, 24] might fail when relaxing the assumption of causal sufficiency. To overcome this gap, one possible solution is to introduce auxiliary information (e.g., instrumental variables or proxy variables [45]). Finally, to remove the unobserved confounding effect, one can allow the partial DAG is given as a prior (bi-directed edges representing the hidden confounders), and a min-max optimization framework [62] might be designed to support the computation of backtracking counterfactuals.

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

## A  Broader Impacts

This paper studies on learning feasible algorithmic recourse without prior causal graph or Structural Causal Models (SCMs) with a novel algorithm, which identifies and constrains the variation of exogenous noise during the recourse process. This advancement in Machine Learning enhances decision-making processes in various fields such as medicine, finance, and social science. For instance, financial institutes often provide recommendations to their clients, by presenting possible and valid suggestions, i.e., recoursed examples, leading to positive decisions, e.g., accepted loan applications. However, the access to prior causal graph or SCMs is difficult for the institute, and our proposed method can bridge such realistic problem when deploying algorithmic recourse models. The limitation of our RATIV algorithm is that it relies heavily on the causal sufficiency assumption, i.e., no unobserved confounders exist. In the case of hidden confounder, our algorithm may not accurately provide valid recoursed samples.

## B  More related work on causally inspired interpretability of ML models

Recently, an extensive body of work has applied causal inference methods to a variety of scenarios relating to the interpretability of ML models, including counterfactual generation [8, 37, 40, 57], counterfactual explanation [12, 25, 27, 38, 39, 55], counterfactual harm [28, 42], interpretable time series prediction [58], causally

interpretable Large Language Models (LLMs) [18, 19, 48], root-cause analysis [4, 17, 20, 49, 50]and data valuation [30], etc.

## C  Difference between traditional counterfactuals and backtracking counterfactuals

*Traditional Counterfactuals.* For a subset of endogenous variables $\tilde{\mathbf{X}} \subseteq \mathbf{X}$ and a realisation $\tilde{\mathbf{x}}$ thereof, the submodel $\mathcal{M}_{\tilde{\mathbf{x}}}$ of $\mathcal{M}$ is the model $\mathcal{M}_{\tilde{\mathbf{x}}} = (\mathbf{X}, \mathbb{F}_{\tilde{\mathbf{x}}}, P_\sigma)$ where $\mathbf{F}_{\tilde{\mathbf{x}}} = \{f_i : V_i \notin \tilde{\mathbf{X}}\} \cup \{\tilde{\mathbf{X}} := \tilde{\mathbf{x}}\}$. The effect of intervening $do(\tilde{\mathbf{X}} = \tilde{\mathbf{x}})$ on $\mathcal{M}$ is given by the submodel $\mathcal{M}_{\tilde{\mathbf{x}}}$. The potential outcome of the recoursed sample $X$ in our paper to action $do(\tilde{\mathbf{X}} = \tilde{\mathbf{x}})$ in world $w = (\mathcal{M}, P_\sigma)$, denoted $\mathbf{X}_{\tilde{\mathbf{x}}}(\sigma)$, is the solution for $\mathbf{X}$ of the set of equations $\mathbb{F}_{\tilde{\mathbf{x}}}$, that is, $\mathbf{X}_{\tilde{\mathbf{x}}}(\sigma) = \mathbf{X}_{\mathcal{M}_{\tilde{\mathbf{x}}}}(\sigma)$. The counterfactual sentence "given that we factually observed $X = \mathbf{X}^F$, how $\mathbf{X}$ would be (in situation $\sigma$), had $\tilde{\mathbf{X}}$ been $\tilde{\mathbf{x}}$" is then interpreted as the equality $\mathbf{X}_{\tilde{\mathbf{x}}}(\sigma) = \mathbf{x}$, where the part "had $\tilde{\mathbf{X}}$ been $\tilde{\mathbf{x}}$" is called the (counterfactual) antecedent. To compute counterfactuals, the three-step procedures include abduction, action, and prediction:

(1) Abduction: Given the evidence, i.e., the factual sample $\mathbf{X}^F$, update $P_\sigma$ by the evidence $\mathbf{X}^F$ to obtain $P_{\sigma|x^F}$.

(2) Action: Modify $\mathcal{M}$ by the action $do\left(\tilde{\mathbf{X}} = \tilde{\mathbf{x}}\right)$ to obtain the submodel $\mathcal{M}_{\tilde{\mathbf{x}}}$.

(3) Prediction: Compute $P_{\mathbf{X}_{do(\tilde{\mathbf{X}}=\tilde{\mathbf{x}})}|x^F}$ with modified $\mathcal{M}_{\tilde{\mathbf{x}}}$ and updated $P_{\sigma|\mathbf{X}^F}$.

*Backtracking Counterfactuals.* Different from traditional counterfactuals, backtracking counterfactuals define interventions as a structure preserving process, i.e., the modifications only happens on the exogenous variables $\sigma$ and the functional relationships $\mathbb{F}$ is preserved [16]. Instead of $\tilde{X}$, the core of backtracking counterfactuals stands on the definition of $\tilde{\sigma}$ with a new conditional term $P(\tilde{\sigma} \mid \sigma)$, i.e., $\tilde{\sigma}$ as the modified exogenous variable and $P(\tilde{\sigma} \mid \sigma)$ as the quantification of the likelihood of each counterfactual world $\tilde{\sigma}$ given factual world $\sigma$ [53]. Then given the factual sample $\mathbf{X}^F$, the probability of backtracking counterfactuals, i.e., $P_B(X^R = x^R, \mathbf{X}^F = \mathbf{X}^F)$ can be computed as follows [53]:

$$\begin{aligned} P_B\left(x^R, \mathbf{X}^F\right) &:= P_B\left(X^R = x^R, \mathbf{X}^F \mathbf{X}^F\right) \\ &= \sum_{(\sigma, \tilde{\sigma})} P_B\left(\sigma, \tilde{\sigma}\right) \mathbf{1}_{\{X^R(\tilde{\sigma})=x^R\}} \mathbf{1}_{\{\mathbf{X}^F(\sigma)=\mathbf{X}^F\}}. \end{aligned} \tag{11}$$

Then the three steps to compute counterfactuals can be formalized to answer the question "given that we factually observed $X = \mathbf{X}^F$, how $\mathbf{X}$ would be (in situation $\sigma$), had $\tilde{\mathbf{X}}$ been $\tilde{\mathbf{x}}$":

(1) Cross-world Abduction: Compute $P(\tilde{\sigma}, \sigma) = P(\sigma)P(\tilde{\sigma} \mid \sigma)$, where the prior $P(\sigma)$ is assumed to be known. Then update $P(\tilde{\sigma}, \sigma \mid \tilde{\mathbf{x}}, x^F)$;

(2) Marginalisation: Marginalise out $\sigma$ to obtain the counterfactual posterior $P_B\left(\tilde{\sigma} \mid \tilde{\mathbf{x}}, x^F\right)$;

(3) Prediction: Use the model $(\mathcal{M}, P_B\left(\tilde{\sigma} \mid \tilde{\mathbf{x}}, x^F\right))$ to predict $X^R$.

We refer more detailed illustration of backtracking counterfactuals to [53]. To be specific, in our paper, as the resulting $X^R$ is deterministically produced by algorithmic recourse models, we judge the conditional term $P_B(\tilde{\sigma} \mid \sigma)$ as a point mass. Besides, we note that the uncertainty from $\sigma$ to $\tilde{\sigma}$ can be supported by considering algorithmic recourse models with uncertainty [24]. Hence, the three steps above, i.e., Cross-world Abduction, Marginalisation and Prediction, can be deduced to the equation (2). Meanwhile, we note that such formalization in (2) coincides with the struture-preserving interventions defined in [4, 16].

## D  Exogeneous-Aggregated Formulation

By recursively resolving the parents in terms of their parents, i.e., using $\mathbf{x}_j := f_j(\mathbf{x}_{pa(j)}, \sigma_j)$, one can easily obtain the form that $\mathbf{x} = f(\sigma)$. For instance, considering the SCM consisting of three variables:

$$
\begin{aligned}
\mathbf{x}_1 &= f_1(\mathbf{x}_2, \mathbf{x}_3, \sigma_1), \\
\mathbf{x}_2 &= f_2(\mathbf{x}_3, \sigma_2), \\
\mathbf{x}_3 &= \sigma_3.
\end{aligned}
\tag{12}
$$

Then one can easily reads that

$$
\begin{aligned}
\mathbf{x}_1 &= f_1(f_2(\sigma_3, \sigma_2), \sigma_3, \sigma_1), \\
\mathbf{x}_2 &= f_2(\sigma_3, \sigma_2), \\
\mathbf{x}_3 &= \sigma_3.
\end{aligned}
\tag{13}
$$

By letting the first output of $f$ to be $f_1(f_2(\sigma_3, \sigma_2), \sigma_3, \sigma_1)$, the second of $f$ to be $\mathbf{x}_2 = f_2(\sigma_3, \sigma_2)$ and the final output of $f$ to be $\mathbf{x}_3 = \sigma_3$, then one can re-write $\mathbf{x}$ as:

$$
\mathbf{x} = f(\sigma).
$$

We note that the crucial part of such form is that the SCM is non-recursive.

## E  Detailed Algorithm

We put detailed illustration of our algorithm in Alg. 1. To be specific, the algorithmic recourse model $\mathcal{M}$, i.e., the CVAE model, serves as both the input and output of our algorithm, as our ultimate goal is to instruct $\mathcal{M}$ to output feasible recoursed samples. We put $\mathcal{M}_l$ as the input to entail that $\mathcal{M}_l$ is untrained and initialized at the beginning, while $\mathcal{M}_l$ will be trained using $\mathcal{D}$ and $\mathcal{D}^A$ in Line 4.

## F  Proofs

Throughout our appendix, we use the subscript $1 \le j \le n$ to index the feature, the subscript $1 \le i \le M$ to index the sample, and the subscript $1 \le k \le K$ to index the order in the conditional exponential distribution.

**Theorem 3.2 (Restated).** *Assume that:*

(a) *The exogenous noise $\sigma$ is conditionally exponential of order $K$ of $y$. Consequently, the conditional probability density of $\sigma$ given $y$ can be written for each feature $1 \le j \le n$:*

$$
p(\sigma_j \mid y) = \frac{Q_j(\sigma_j)}{Z_j(y)} \exp\left[\sum_{k=1}^{K} \tilde{q}_{jk}(\sigma_j)\lambda_{jk}(y)\right],
\tag{14}
$$

---

**Algorithm 1** Illustrations of AR-Nuc and AR-L2

**Require:** The collected observational dataset $\mathcal{D} = \{\mathbf{x}_i^F, y_i\}_{i=1}^M$, the algorithmic recourse model (CVAE) $\mathcal{M}$, the regression model $\mathcal{M}_l = \{\psi, \phi\}$, the size of input features n, the batch size of $\mathcal{M}$ as $M_b$.

**Ensure:** The trained model $\mathcal{M}$.

1: **Extracting the exogenous representations**:
2: Randomly shuffle $\{y_i\}_{i=1}^M$ and obtain the permuted $\widehat{y}$;
3: Construct the augmented dataset $\mathcal{D}^A = \{\mathbf{x}_i^F, \widehat{y}_i\}_{i=1}^M$;
4: Optimize the regression model $\mathcal{M}_l$ as in (3) by discriminating between $\mathcal{D}$ and $\mathcal{D}^A$.
5: **Training the algorithmic recourse model**:
6: Arrive the latent representation $\{\mathbf{z}_i\}_{i=1}^{M_b}$ through the encoder from the input $\{\mathbf{x}_i^F\}_{i=1}^{M_b}$ with target labels $\{y_i'\}_{i=1}^{M_b}$;
7: By sampling from $\{\mathbf{z}_i\}_{i=1}^{M_b}$, compute the reconstructed $\{\mathbf{x}_i^R\}_{i=1}^{M_b}$ from the decoder with $\{y_i'\}_{i=1}^{M_b}$;
8: Compute the original objective $\mathcal{L}_{\text{ori}}$ of $\mathcal{M}$ with $\{\mathbf{z}_i, \mathbf{x}_i^R, \mathbf{x}_i^F, y_i, y_i'\}_{i=1}^{M_b}$;
9: **AR-Nuc**: Optimize the total objective as $\mathcal{L}_{\text{ori}} + \alpha_{nuc}\mathcal{L}_{\text{nuc}}$.
10: **AR-L2**: Optimize the total objective as $\mathcal{L}_{\text{ori}} + \alpha_{mag}\mathcal{L}_{\text{mag}}$.

---

*where $Q_j, Z_j, \tilde{q}_{jk}$ and $\lambda_{jk}$ as scalar-valued functions. Meanwhile, for each $j$, the sufficient statistics $\tilde{q}_{jk}$ are assumed linearly independent over $k$.*

(b) *There exists $nk + 1$ realizations of $y$ as $\{y\}_{l=0}^{nk}$ such that the matrix with size $nk \times nk$:*

$$
\mathbf{L} = \begin{pmatrix}
\lambda_{11}(y_1) - \lambda_{11}(y_0), \ldots, \lambda_{11}(y_{nk}) - \lambda_{11}(y_0) \\
\vdots \\
\lambda_{nk}(y_1) - \lambda_{nk}(y_0), \ldots, \lambda_{nk}(y_{nk}) - \lambda_{nk}(y_0)
\end{pmatrix}
\tag{15}
$$

*is invertible.*

(c) *The trained (deep) logistic regression system in (3) has the universal approximation capability to distinguish $\mathcal{D}$ from $\mathcal{D}^A$.*

*Then, in the case of infinite samples, the representations $\phi^\theta(\mathbf{x})$ identifies $\sigma$ up to a linear transformation of point-wise statistics $\tilde{\mathbf{q}}$:*

$$
\tilde{\mathbf{q}}(\sigma) = \mathbf{A}\phi_\theta(\mathbf{x}) + \mathbf{b},
\tag{16}
$$

*here $\mathbf{A}$ and $\mathbf{b}$ are fixed but unknown matrices.*

PROOF FOR THEOREM 3.2. Overall, our techniques in this proof are inspired by the previous results in [15]. First, with the properties that $\sigma_{j1}$ is statistically dependent on $y$, but conditionally independent of the other $\sigma_{j2}$, we have the following expression:

$$
\log p(\sigma \mid y) = \sum_{j=1}^{n} q_j(\sigma_j, y).
\tag{17}
$$

Furthermore, based on previous well-known results [11], the universal approximation capability assumption in our theorem implies that the regression function $r$ will equal the difference of the log-densities in the two classes (namely $\mathcal{D}$ and $\mathcal{D}^A$):

$$
\begin{aligned}
\sum_{j=1}^{n} \psi_j\left(\phi_j(\mathbf{x}), y\right) = {} & \log p(\sigma, \mathbf{u}) + \log|det\mathbf{J}g(\mathbf{x})| - \log p(\sigma) \\
& - \log p(y) - \log|det\mathbf{J}g(\mathbf{x})|,
\end{aligned}
\tag{18}
$$

where the term $det\mathbf{J}g(\mathbf{x})$ refers to the determinant of the Jacobian matrix of $g$, and the equality holds due to the fact that the $p(\sigma, y) = p(\sigma)p(y)$ in $\mathcal{D}^A$. Meanwhile, based on the conditional-exponential assumption, the left side of the above equation can be simplified into the following expression:

$$\sum_j \log Q_j\left(\sigma_j\right) + \left[\sum_k \tilde{q}_{jk}\left(\sigma_j\right)\lambda_{jk}(y)\right] - \log Z_j(y) - \log p(\sigma). \quad (19)$$

Consequently, a linear solution of $\sum_{j=1}^n \psi_j\left(\phi_j(\mathbf{x}), y\right)$ can be written as follows:

$$\sum_{jk} \tilde{\phi}_{jk}(\mathbf{x})v_{jk}(y) + s(\mathbf{x}) + t(\mathbf{u}), \quad (20)$$

where

$$\begin{aligned} \tilde{\phi}_{jk}(\mathbf{x}) &= \tilde{q}_{jk}(\sigma_j) \\ v_{jk}(y) &= \lambda_{jk}(y) \\ s(\mathbf{x}) &= \sum_j \log Q_j\left(\sigma_j\right) - \log p(\sigma) \\ t(y) &= \sum_j -\log Z_j(y), \end{aligned} \quad (21)$$

where the representations $\tilde{\phi}(\mathbf{x})$ identifies exactly the $\tilde{q}(\sigma)$ in this special solution. Moreover, we show that the above solution for the regressor is the only solution up to the $\mathbf{A}, \mathbf{b}$ given in the theorem (namely, $\tilde{\phi}$ identifies $\tilde{q}$ up to a linear transformation). To this end, we collect the following equations for the points $y_{l=1}^{nk+1}$ in the assumption 2 in our theorem:

$$\sum_{jk} \tilde{\phi}_{jk}(\mathbf{x})v_{jk}(y_l) + s(\mathbf{x}) + t(\mathbf{u})$$
$$= \sum_j \log Q_j\left(\sigma_j\right) + \left[\sum_k \tilde{q}_{jk}\left(\sigma_j\right)\lambda_{jk}(y_l)\right] - \log Z_j(y) - \log p(\sigma),$$
$$(22)$$

then the following matrix expression is obtained:

$$\mathbf{W}^T\tilde{\phi}(\mathbf{x}) = \mathbf{L}^T\tilde{\mathbf{q}}(\sigma) - \mathbf{z} + \mathbf{1}\left[\sum_j \log Q_j\left(\sigma_j\right) - q_0(\sigma) - a(\mathbf{x})\right], \quad (23)$$

where $\mathbf{W} \in \mathcal{R}^{nk\times(nk+1)}$ is the matrix expression of the vectors $\mathbf{W}(y_l)$ $(1 \le l \le nk)$, $\mathbf{L} \in \mathcal{R}^{nk\times(nk+1)}$ is the matrix form of $\lambda_{jk}(y_l)$ with $j*k$ as the row index and $l$ as the column index, $\tilde{\mathbf{q}}(\sigma) \in \mathcal{R}^{nk}$ is the collection of $\tilde{q}_{jk}\left(\sigma_j\right)$, $\tilde{\phi}(\mathbf{x}) \in \mathcal{R}^{nk}$ is the representation vector, $\mathbf{z} \in \mathcal{R}^{nk+1}$ is the collections of all $t(y_l) + \sum_j \log Z_j(y_l)$ for different $l$, and $\mathbf{1} \in \mathcal{R}^{nk+1}$ is a vector of ones. Moreover, we subtract the first row of the above equation from its rest rows, and derive the following equation:

$$\widehat{\mathbf{W}}^T\tilde{\phi}(\mathbf{x}) = \widehat{\mathbf{L}}^T\tilde{\mathbf{q}}(\sigma) - \widehat{\mathbf{z}}, \quad (24)$$

where $\widehat{\mathbf{W}}$ and $\widehat{\mathbf{L}}$ are differences of the rows of $\mathbf{W}$ and $\mathbf{L}$ (and likewise for $\widehat{\mathbf{z}}$). Finally, since the matrix $\widehat{\mathbf{L}}$ coincides with invertible assumption (b) in our theorem, we obtain the identification results as follows:

$$\mathbf{A}\tilde{\phi}(\mathbf{x}) = \tilde{\mathbf{q}}(\sigma) - \mathbf{b}, \quad (25)$$

where $\mathbf{A} = \widehat{\mathbf{L}}^{-1}\widehat{\mathbf{W}}$ and $\mathbf{b} = \widehat{\mathbf{L}}^{-1}\widehat{\mathbf{z}}$. Notably, the unknown matrices $\mathbf{A}$ and $\mathbf{b}$ only depend on the support points $y$. $\square$

PROOF FOR THEOREM 3.3. First, we list the expression of $\mathbf{H}^\sigma$ and $\mathbf{H}^0$ and $\mathbf{H}$ for convenience:

$$\begin{cases} \mathbf{H}^\sigma = \{\tilde{\mathbf{q}}(\sigma_1^F) - \tilde{\mathbf{q}}(\sigma_1^{CF}), \tilde{\mathbf{q}}(\sigma_2^F) - \tilde{\mathbf{q}}(\sigma_2^{CF}), \ldots, \tilde{\mathbf{q}}(\sigma_{M_b}^F) - \tilde{\mathbf{q}}(\sigma_{M_b}^{CF})\}, \\ \mathbf{H}^0 = \{\sigma_1^F - \sigma_1^{CF}, \sigma_2^F - \sigma_2^{CF}, \ldots, \sigma_{M_b}^F - \sigma_{M_b}^{CF}\}, \\ \mathbf{H} = \{\phi(\mathbf{x}_1^F) - \phi(\mathbf{x}_1^{CF}), \phi(\mathbf{x}_2^F) - \phi(\mathbf{x}_2^{CF}), \ldots, \phi(\mathbf{x}_{M_b}^F) - \phi(\mathbf{x}_{M_b}^{CF})\}, \end{cases}$$

Combinin with the fact that $AH = H^\sigma$, $\tilde{\mathbf{q}}$ is point-wise, we have $\mathcal{L}_0(H^0) \le \mathcal{L}_0(H^\sigma)$. Then we have to prove that $\epsilon\mathcal{L}_0(H^\sigma) \le \mathcal{L}_0(H) \le cn\mathcal{L}_{nuc}(H)$. Based on the assumption that the transformed exogenous noise is bounded, i.e., $\epsilon \le \|H_{ij}^\sigma\|$, we have $\epsilon\mathcal{L}_0(H^\sigma) \le \mathcal{L}_0(H)$. Then by Cauchy's Inequality and $AH = H^\sigma$, we assume that the matrix $A$ is bounded by some constant $c$:

$$\begin{aligned} \mathcal{L}_1(H^\sigma) &= \max_{j\in[M_b]} \sum_{i\in[n]} |\sum_j A_{ij}H_j| \\ &\le \max_{j\in[M_b]} \sum_{i\in[n]} \sum_j |A_{ij}||H_j| \\ &\le cn\mathcal{L}_1(H), \end{aligned} \quad (26)$$

By invoking the Cauchy's inequality again, we have:

$$\begin{aligned} cn\mathcal{L}_1(H) &= cn \max_{j\in[b]} \sum_{i\in[n]} |H_{ji}| \\ &\le cn\sqrt{n} \max_{j\in[b]} \sqrt{\sum_i H_{ij}^2} \\ &= cn\sqrt{n}\text{Trace}\sqrt{H^T H}, \end{aligned} \quad (27)$$

where the term $\text{Trace}\sqrt{H^T H}$ equals to $\mathcal{L}_{nuc}(H)$. Then our claim follows.

$\square$

PROOF FOR THEOREM 3.4. First, we show that the term $\|\tilde{\mathbf{q}}(\sigma^F) - \tilde{\mathbf{q}}(\sigma^{CF})\|_2$ is bounded by $\|\phi(\mathbf{x}^F) - \phi(\mathbf{x}^{CF})\|_2$ as follows:

$$\begin{aligned} \|\tilde{\mathbf{q}}(\sigma^F) - \tilde{\mathbf{q}}(\sigma^{CF})\|_2 &= \|\mathbf{A}\phi(\mathbf{x}^F) - \phi(\mathbf{x}^{CF})\|_2 \\ &\le \|\mathbf{A}\|\|\phi(\mathbf{x}^F) - \phi(\mathbf{x}^{CF})\|_2, \end{aligned} \quad (28)$$

where the first equality is due to the results of identification, and the second inequality is due to the definition of the norm of the operator $\mathbf{A}$. Moreover, as $\mathbf{A}$ maps between finite-dimensional Hibert spaces and $\mathbf{A}$ is a continuous operator, $\mathbf{A}$ itself is bounded (e.g., $\|\mathbf{A}\| \le C$ holds). Meanwhile, recalling our assumption that $\tilde{\mathbf{q}}$ is a bi-lipschitz function, we have:

$$K_1\|\sigma^F - \sigma^{CF}\|_2 \le \|\tilde{\mathbf{q}}(\sigma^F) - \tilde{\mathbf{q}}(\sigma^{CF})\|_2 \le K_2\|\sigma^F - \sigma^{CF}\|_2, \quad (29)$$

where $K_1$ and $K_2$ are Lipschitz constants. Notably, such assumpion implies that the variation of $\tilde{\mathbf{q}}$ is compactly correlated to that of $\sigma$, which is common for exponential families [14]. Hence, $\|\sigma^F - \sigma^{CF}\|_2$ is governed by $\|\phi(\mathbf{x}^F) - \phi(\mathbf{x}^{CF})\|_2$:

$$\|\sigma^F - \sigma^{CF}\|_2 \le \frac{1}{K_1}\|\mathbf{A}\|\|\phi(\mathbf{x}^F) - \phi(\mathbf{x}^{CF})\|_2, \quad (30)$$

where minimizing $\|\phi(\mathbf{x}^F) - \phi(\mathbf{x}^{CF})\|_2$ is enough to constrain $\|\sigma^F - \sigma^{CF}\|_2$. $\square$

# G Experimental Details

*Details on Implementation of baselines.* The CFVAE baseline serves as the underlying algorithmic recourse model for our AR-Nuc and AR-L2 methods. We follow the implementations in [31], using Multi-layer-perception (MLP) layers to estimate $\mu_{y'}$ and $\sigma_{y'}^2$ in the encoder branches. The black box ML model $h$ is also an MLP classifier. We use the Adam optimizer [2] with an initial learning rate of 0.01 for $h$ and $\mathcal{M}$. The batch size $M_b$ is set to 64 in all our experiments. The original implementations of AR-SCM and CFVAE in Pytorch by [31] are publicly available[5]. The CEM[6] and CFA[7] methods are also open-source on GitHub.

*Computing Resources.* All of our experiments are conducted on a GPU server with 8 Nvidia 3090, Pytorch 1.12, Cuda 12.1.

*Details on Dataset.* We then detail the simulation on the German Loan dataset, with the same protocols in [21] as follows:

$$
\begin{aligned}
&G : U_G, && U_G \sim \text{Bernoulli}(0.5) \\
&A := -35 + U_A, && U_A \sim \text{Gamma}(10, 3.5) \\
&E := -0.5 + \left(1 + e^{-\left(-1 + 0.5G + \left(1 + e^{-0.1A}\right)^{-1} + U_E\right)}\right)^{-1}, && U_E \sim \mathcal{N}(0, 0.25) \\
&L := 1 + 0.01(A - 5)(5 - A) + G + U_L, && U_L \sim \mathcal{N}(0, 4) \\
&D := -1 + 0.1A + 2G + L + U_D, && U_D \sim \mathcal{N}(0, 9) \\
&I := -4 + 0.1(A + 35) + 2G + GE + U_I, && U_I \sim \mathcal{N}(0, 4) \\
&S := -4 + 1.5\mathbb{I}_{\{I>0\}}I + U_S, && U_S \sim \mathcal{N}(0, 25).
\end{aligned}
\tag{31}
$$

Meanwhile, we generate the class label $y$ following [24]:

$$
y \sim \text{Bernoulli}\left(\left(1 + e^{-0.3(-L - D + I + S + IS)}\right)^{-1}\right).
\tag{32}
$$

Besides, we provide details on the sample number for each dataset. For the synthetic dataset and semi-synthetic German Load dataset, we set $M = 10000$ as the number of samples. For the real-world Diabetes dataset, we have $M = 768$ samples. Such variation on the samples size also verifies that our methods does not rely on huge data samples. To report the out-of-sample prediction results, we randomly split the each dataset into the training/testing domains with ratio as 0.7/0.3.

*Details on generating the high-dimensional Dataset.* The rationale behind employing synthetic data is twofold: (a) most widely used, realistic datasets possess relatively small feature dimensions; (b) real-world data with high dimensions lacks an underlying SCM, rendering it difficult to evaluate the feasibility.

We have augmented our research by incorporating additional experiments involving a synthetic dataset in our study. The synthetic dataset is designed with a feature dimension of 80. The rationale behind employing synthetic data is twofold: (a) most widely used, realistic datasets possess relatively small feature dimensions; (b) real-world data lacks an underlying structural causal model (SCM), rendering it infeasible to verify whether generated explanations align with the SCM model. Specifically, we extend the synthetic setting in our paper to encompass a high-dimensional scenario with 80 dimensions:

$$
\begin{aligned}
\mathbf{x}_1 &\sim N(\mu_1, \sigma_1); \\
\mathbf{x}_2 &\sim N(\mu_2, \sigma_2); \\
\mathbf{x}_3 &\sim N(\mu_1, \sigma_1); \\
\mathbf{x}_4 &\sim N(\mu_2, \sigma_2); \\
&\cdots, \\
\mathbf{x}_8 &\sim N(\mu_2, \sigma_2); \\
\mathbf{x}_9 &\sim N\left(k_1 * (\mathbf{x}_1 + \mathbf{x}_2 + \mathbf{x}_3 + \mathbf{x}_4)^2 + b_1, \sigma_3\right); \\
\mathbf{x}_{10} &\sim N\left(k_1 * (\mathbf{x}_5 + \mathbf{x}_6 + \mathbf{x}_7 + \mathbf{x}_8)^2 + b_1, \sigma_3\right).
\end{aligned}
\tag{33}
$$

In order to augment the original dataset $X^F$ and create a more complex structure, additional variables $\mathbf{x}_{i*10+1}$ to $\mathbf{x}_{i*10+10}$ are generated for $1 \leq i \leq 7$, following the same procedure as $i = 0$. Subsequently, a random permutation is applied to shuffle the variables $\mathbf{x}1$ to $\mathbf{x}80$. This permutation aims to challenge the preservation of the original structure in $X^F$.

To ensure the integrity of the modified dataset, a feasibility check is performed. Specifically, for each sample, the following conditions are validated: - If $\mathbf{x}_i, \mathbf{x}_{i+1}, \mathbf{x}_{i+2}, \mathbf{x}_{i+3}$ increase, then $\mathbf{x}_{i+9}$ must also increase for $i = 10k + 1$, where $0 \leq k \leq 7$. - If $\mathbf{x}_i, \mathbf{x}_{i+1}, \mathbf{x}_{i+2}, \mathbf{x}_{i+3}$ decrease, then $\mathbf{x}_{i+9}$ must also decrease for $i = 10k + 1$, where $0 \leq k \leq 7$. - Similarly, for each sample, if $\mathbf{x}_i, \mathbf{x}_{i+1}, \mathbf{x}_{i+2}, \mathbf{x}_{i+3}$ increase, $\mathbf{x}_{i+5}$ should also increase, and if they decrease, $\mathbf{x}_{i+5}$ should also decrease, for $i = 10k + 5$, where $0 \leq k \leq 7$.

*Details on the Regression System.* Moreover, we perform extra experiments to illustrate the behaviour of our regression system for extracting the exogenous representations. To be specific, we report the training process on the Diabetes dataset in Figure 6, where the convergent training loss indicates that the model indeed achieves nearly the universal approximation capability (which is critical for identifying the exogenous noise in our theorem).

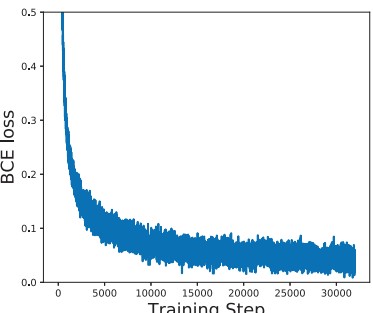

**Figure 6: The training curve of our regression system.**

Received 20 February 2007; revised 12 March 2009; accepted 5 June 2009

---

[5]https://github.com/divyat09/AR-feasibility
[6]https://github.com/IBM/Contrastive-Explanation-Method
[7]https://github.com/amirhk/recourse

