# OpenReview forum: "Learning Feasible Causal Algorithmic Recourse: A Prior Structural Knowledge Free Approach"
_ACM.org/TheWebConf/2025/Conference — WWW 2025 Poster_

### Official Review · Reviewer_qnuq · 2024-11-21

**Novelty:** 5
**Technical Quality:** 4

**Review:**

The paper addresses the challenges in **algorithmic recourse (AR)**, which involves providing actionable recommendations to individuals affected by machine learning (ML) decisions. Traditional AR methods often fail to ensure feasibility in real-world applications because they disregard the causal relationships among input features. The paper aims to propose a method for causal algorithmic recourse that does not rely on predefined causal graphs or SCMs.

---

**strength**

1. The method reformulates algorithmic recourse as a structure-preserving intervention.
2. The approach of using exogenous noise identification is innovative, providing theoretical guarantees while maintaining computational feasibility.
3. Two models are proposed: AR-L2 (constrains the magnitude of exogenous noise variations) and AR-Nuc (imposes sparsity constraints using nuclear norms).

---

**Limitations**

1. **Optimization Challenges**

The paper offers limited analysis regarding the optimization process and the interaction of its multiple loss components.

This method incorporates several loss constraints to guide the training process, including reconstruction loss from the CVAE, prediction change loss, constraints on exogenous noise variations, and KL divergence terms. The interplay between these loss terms could potentially introduce training instabilities, which are not explored in detail.


2. **Explicit Causal Relationships**

Although the proposed method does not require explicit causal graphs, it does rely on the assumption that exogenous noise is conditionally exponential. Verifying whether exogenous noise adheres to this conditional exponential distribution in the provided dataset—or in real-world data—can be highly challenging.

3. **Structural Knowledge**

The method's feasibility evaluation seems to still depends on predefined constraints. For instance, in the German Loan dataset, rules such as *"an increase in loan amount leads to an increase in loan duration"* or *"an increase in age corresponds to an increase in income"* are examples of prior structural knowledge. This raises an important question: how do these predefined constraints differ from the broader notion of "prior structural knowledge"?

4. **Overall Framework**

The paper primarily emphasizes theoretical analysis and conceptual solutions to the identified problems, but it provides limited detail regarding the model's implementation, as well as the specific changes and/or improvements compared to prior work from the implemental perspective.

**Questions:**

1. How do you tune the different term in the loss function? The adaptive alignment of these weights may yield significant improvements in performance, but further clarification is needed on how such tuning is handled.

2. How can the assumption of conditional exponentiality for exogenous noise be effectively tested or validated within the provided dataset?

3. How do these predefined constraints differ from the broader notion of "prior structural knowledge"? Additionally, if predefined rules are employed during experiments, does this compromise the claimed contribution of working without prior structural knowledge?

4. Please clarify the differences between the proposed framework with CVAE and CFVAE.

**Reviewer Confidence:**

2: The reviewer is willing to defend the evaluation, but it is likely that the reviewer did not understand parts of the paper

**Scope:**

3: The work is somewhat relevant to the Web and to the track, and is of narrow interest to a sub-community

---

### Official Review · Reviewer_92T5 · 2024-11-22

**Novelty:** 5
**Technical Quality:** 5

**Review:**

Prior works on causal Algorithmic Recourse (AR) require prior structural causal knowledge from prior causal graph or the whole structural causal model. This work avoids this requirement by proposing to approximate and constrain the variation of the exogenous noise variables  by formulating the generation of AR as the structure-preserving intervention. The authors proposed two methods AR-L2, to reduce the magnitude of variations in exogenous representations, and AR-Nuc, to reduce the number of adjustment variations. The proposed methods perform better than selected baselines with partial or no prior causal knowledge and approach the performance of the method with the whole SCM model as a prior (oracle).


**Pros:**
- (P1) The novel method tackle causal algorithmic recourse without requiring Prior Causal Graph or Prior SCM.
- (P2) The experiment results sufficiently show how well the proposed method perform on preserving the causal relationship and feasibility enhancement.

**Cons:**
- (C1) The source code and data should be provided for reproducibility.
- (C2) While the experiment settings are details. The discussion of evaluation can be enhanced further by restructuring the paragraphs into subsections focusing on answering main research questions.
- (C3) A case study on a real dataset can be presented for showcase the quality of the proposed method.

Minor typos:
- Line 106: eailer -> earlier
- Line 414: learned in (16)  -> learned in (4)
- Line 490, 494: nu-clear norm -> nuclear norm
- Line 688, 690: \sigma^{2}y' -> \sigma^{2}_{y'}
- Line 689: 5 MLP layers
- Line 693: 2 MLP layers

**Questions:**

- (Q1) Can we incorporate both L_nuc and L_mag into the overall objective? Please elaborate more.
- (Q2) What are the the statistical test results? (e.g., AR-L2 vs. CFVAE, AR-L2 vs. Oracle)

**Reviewer Confidence:**

3: The reviewer is confident but not certain that the evaluation is correct

**Scope:**

3: The work is somewhat relevant to the Web and to the track, and is of narrow interest to a sub-community

---

### Official Review · Reviewer_HzGz · 2024-11-30

**Novelty:** 5
**Technical Quality:** 5

**Review:**

Traditional causal recourse methods rely on complete causal graphs, which are often difficult to obtain in real-world scenarios. This paper leverages non-linear Independent Component Analysis (ICA) to propose AR-L2 and AR-Nuc, enabling the generation of feasible Algorithmic Recourse without requiring prior causal knowledge.
### Advantages of the Paper
1. This paper proposes a method that does not require prior causal knowledge, lowering the barrier to applying causal recourse.
2. It introduces two effective recourse generation methods (AR-L2 and AR-Nuc).
3. The effectiveness of the methods is validated through real-world datasets and simulated datasets.
### Limitations of the Paper
1. The explanation of Theorem 4.2 is not sufficiently clear.
2. The writing of the paper needs improvement.

**Questions:**

1. Regarding Limitation 1, specifically the question about "Identification of the Exogenous Noise," the paper trains a deep learning model to identify exogenous variables $\sigma$. It does so by randomly permuting labels to allow the model to learn. However, it is unclear how the model ensures that it learns only the exogenous variables $ \sigma$ while excluding the influence of the causal parent values $x_{\text{pa}(j)}$
2. There are ambiguities in some sentences, such as “Alternatively, we propose that identifying another core term in (2) and (2),” and some references to equations, like (16), have been placed in the appendix. It would be better to reorganize and include them in the main text.

**Reviewer Confidence:**

2: The reviewer is willing to defend the evaluation, but it is likely that the reviewer did not understand parts of the paper

**Scope:**

3: The work is somewhat relevant to the Web and to the track, and is of narrow interest to a sub-community

---

### Official Review · Reviewer_XHwG · 2024-12-02

**Novelty:** 4
**Technical Quality:** 4

**Review:**

This paper proposes a new way to generate feasible algorithmic recourse without needing prior causal graphs or full causal models, which are often unavailable in real-world scenarios. Instead, the authors use structure-preserving interventions and non-linear ICA to handle causal relationships, ensuring that the generated counterfactuals are realistic and actionable. Their methods (AR-L2 and AR-Nuc) were tested on multiple datasets, showing better performance in preserving causal links and altering predictions effectively.

## Strengths

1. The approach works without needing existing causal graphs or models, which makes it much easier to apply in real-world situations.
2. By using structure-preserving interventions, the method keeps causal relationships intact, leading to more practical, realistic recommendations.
3. The methods (AR-L2 and AR-Nuc) were tested on diverse datasets, consistently showing better feasibility and accuracy compared to existing approaches.

## Weakness:

1. The method relies heavily on the assumption of causal sufficiency, which limits its applicability in real-world scenarios where unobserved confounders are often present. This assumption may reduce the robustness and generalizability of the proposed framework.
2. Computational complexity is a concern, especially for high-dimensional data. The current method does not sufficiently address efficiency optimization, which might hinder its use in practical, large-scale, or real-time applications.

**Questions:**

1. How does the proposed framework perform when causal sufficiency is violated, and unobserved confounders exist? Are there strategies, such as proxy variables, that could mitigate these limitations and ensure the reliability of counterfactual recommendations?

2. Given the computational demands of the proposed approach, I wonder is there a potential for developing efficient approximations or distributed computing solutions to enhance scalability and make the method feasible for real-time use cases?

**Reviewer Confidence:**

3: The reviewer is confident but not certain that the evaluation is correct

**Scope:**

3: The work is somewhat relevant to the Web and to the track, and is of narrow interest to a sub-community